# Three-Dimensional Vibration Analysis of a Functionally Graded Sandwich Rectangular Plate Resting on an Elastic Foundation Using a Semi-Analytical Method

**DOI:** 10.3390/ma12203401

**Published:** 2019-10-17

**Authors:** Jie Cui, Taoran Zhou, Renchuan Ye, Oleg Gaidai, Zichao Li, Shenghui Tao

**Affiliations:** School of Naval Architecture and Ocean Engineering, Jiangsu University of Science and Technology, Zhenjiang 212003, China; cuijie2006@hotmail.com (J.C.); zhoutaoran@163.com (T.Z.); oleg.gaidai@yahoo.com (O.G.); 17751192059@163.com (Z.L.); WYtaoshenghui@163.com (S.T.)

**Keywords:** three-dimensional plate vibration, Ritz method, general boundary conditions, elastic foundation, functionally graded sandwich plate

## Abstract

The three-dimensional vibration of a functionally graded sandwich rectangular plate on an elastic foundation with normal boundary conditions was analyzed using a semi-analytical method based on three-dimensional elasticity theory. The material properties of the sandwich plate varied with thickness according to the power law distribution. Two types of functionally graded material (FGM) sandwich plates were investigated in this paper: one with a homogeneous core and FGM facesheets, and another with homogeneous panels and an FGM core. Various displacements of the plates were created using an improved Fourier series consisting of a standard Fourier cosine series along with a certain number of closed-form auxiliary functions satisfying the essential boundary conditions. The vibration behavior of the FGM sandwich plate, including the natural frequencies and mode shapes, was obtained using the Ritz method. The effectiveness and accuracy of the suggested technique were fully verified by comparing the natural frequencies of sandwich plates with results from investigations of other functionally graded sandwich rectangular plates in the literature. A parametric study, including elastic parameters, foundation parameters, power law exponents, and layer thickness ratios, was performed, and some new results are presented.

## 1. Introduction

In previous decades, functionally graded materials (FGMs) have been widely used in aviation, nuclear energy, and other fields because of their special physical characteristics under high temperatures. The development of pure functionally gradient materials and FGM sandwich plates has been aimed at easing large interfacial shear stresses. Compared with pure functionally graded materials, there are relatively few studies on FGM sandwich structures. Thus, further research on these structures will be helpful for scientists.

Mechanical properties of sandwich plates with a homogeneous and functionally graded material core have been investigated in the area of long-span lightweight structures [1,2,3,4]. Theoretical models, such as shear deformation plate theory, energy method, finite element method, and three-dimensional elastic theory, have been used to investigate the mechanical behavior of FGM structures. To account for specific deformations and stresses, many specific plate and shell models for sandwich/laminate/FGM plates have been developed. Four theoretical models are often used to analyze the complex mechanical behavior of sandwich/laminate/FGM plates: (1) equivalent single-layer theory (ESLT), which can be divided into three main categories, namely classical plate theory (CPT) [5,6], first-order shear deformation theory (FSDT) [7,8], and higher-order shear deformation theory (HSDT) [9,10,11]; (2) refined higher-order shear deformation theory (RHSDT) [12,13,14,15,16]; (3) 3D elasticity theory [17,18]; and (4) layer-wise theory (LWT) [19,20,21,22]. In ESLT, the sandwich plate is analyzed in terms of displacement functions related to a single reference surface. Exact 3D elasticity benchmark solutions for the staticity, vibrations, and dynamics of sandwich plates have been presented by researchers [9,12]. The unified formulation proposed by Carrera [23,24] provides a procedure to describe and implement numerous plate/shell theories and finite elements in a unified manner by referring to a few fundamental nuclei. Individual layers in LWT are considered separately as independent displacement fields with compatibility enforced by appropriate interface compatibility constraints. In addition to ESLT and LWT, other theories exist; for example, simplified equivalent single-layer theory (SESLT) [25,26,27], global–local higher-order theory [28,29], and mixed layer-wise theory (MLWT) [30,31]. SESLT divides transverse displacement into bending and shear parts and reduces the displacement field unknowns. By contrast, MLWT is formulated on the basis of the displacement and transverse stress at interfaces.

The study of vibration characteristics of sandwich rectangular plates with an isotropic or FGM core has made some improvement in recent years. Guided by Reddy’s third-order shear deformation theory, Moita et al. [32] researched the vibration characteristics of material sandwich structures with passive damping by utilizing the finite element method. Taking into account high-order shear deformation theory (HSDT), Thai et al. [33] studied various properties of sandwich FGM plates, including free vibration, bending, and buckling, by applying a moving Kriging mesh-free method. Tounsi et al. [34] proposed a refined trigonometric shear deformation theory (RTSDT) produced using functional gradient sandwich plates under classical boundary conditions. In the analysis of free and forced vibrations on an elastic foundation, Trinh and Kim [35] presented analytical closed-form solutions for thin FGM sandwich shells with a resting double curvature. Aiming at a dynamic, as well as static, analysis of FGM sandwich plates, Pandey and Pradyumna [36,37] extended the higher-order hierarchical finite element formula. According to the theory of higher-order shear deformation plates, Daikh and Megueni [38] applied closed-form analytical solutions to analyze the thermal buckling of functionally graded sandwich plates with simple support boundary conditions. Using a meshless technique, Neves et al. [39] investigated the characteristics of free vibration and analyzed the buckling of functionally graded sandwich plates on the condition that the boundary is simply supported under the guidance of a theory called quasi-3D higher-order shear deformation. Furthermore, using four-variable refinement plate theory as a guideline, Bourada et al. [40] determined the thermal buckling characteristics of FGM sandwich plates. In light of the theory of hyperbolic shear deformation, Houari et al. [41] explored buckling, as well as the free vibration, of thick FGM sandwich plates under the conditions of an ordinary supported boundary. Using 3D linear elastic theory, Li et al. [42] adopted the theoretical content of the Ritz method to study the vibration characteristics of rectangular plates. Jalali et al. [43] used a pseudo-spectral method to study the thermal stability features of FGM sandwich plates subject to a uniform temperature. Houari et al. [44] discussed the thermoelastic bending deformation of FGM sandwich plates under simply supported boundary conditions in detail according to the two-variable refined plate theory and analyzed stresses. In order to explore the vibration characteristics of FGM sandwich plates, Thai et al. [45,46], Hadji and Tounsi et al. [47,48] proposed an innovative simple modified plate theory, which proved to be in good agreement with experiments. Neves et al. [49] conducted a static analysis of FGM sandwich plates under the guidance of a meshless method with a radial basis function assignment. For more related research results on sandwich beams and sandwich shells, see References [50,51,52,53,54,55,56,57,58,59].

The above review shows that most existing studies on sandwich rectangular plates are confined to two-dimensional elastic theory, including the first-order shear deformation theory (FSDT) and the high-order shear deformation theory (HSDT). So far, only Li’s [42] work has been in accordance with the theory of three-dimensional elasticity. Besides, most researchers limited their study to classical boundary conditions, e.g., simply supported boundary conditions. Based on existing research results, the main objective of this study was to establish a three-dimensional sandwich rectangular plate vibration analysis model with general boundary conditions and an elastic foundation. The Pasternak model was used to model the elastic foundation. Also, general boundary conditions were realized using three sets of linear springs uniformly distributed at the edges. The theoretical formulation was achieved based on the Ritz and three-dimensional elasticity theories. Throughout this paper, the authors aim to enrich existing research results of sandwich rectangular plates and provide reference data for future follow-up studies.

## 2. Theoretical Formulations

### 2.1. Geometrical Configuration

Figure 1 gives a schematic diagram of a three-dimensional sandwich rectangular plate on an elastic foundation under general boundary conditions. Figure 1a utilizes the Cartesian coordinate system (*x*, *y*, *z*), with the origin (0, 0, 0) at the bottom of a rectangular plate. The letter *a*, *b*, and *h* denote the length, width, and thickness of a rectangular plate in the *x*-, *y*-, and *z*-directions, respectively. Figure 1b shows general boundary conditions using three sets of linear springs (*k_u_*, *k_v_*, *k_w_*). Three elastic layers make up the sandwich plate; *h*_1_ = 0, and *h*_2_, *h*_3__,_ and *h*_4_ = *h* represent vertical coordinates from the bottom to the top, including the two middle interfaces. For the sake of simplicity, a combination of three numbers (1–0–1, 2–1–2, etc.) are used to represent the thickness ratio of each layer from the bottom to the top.

In this paper, two common sandwich structures are used to describe the material characteristics of three-dimensional sandwich rectangular plates, as shown in Figure 2. Two kinds of sandwich structures are denoted as type A and type B. Type A contains an FGM face sheet and a homogeneous core; type B contains a homogeneous face sheet and an FGM core. The volume fraction *V*_1_ of an FGM sandwich rectangular plate is defined as:(1)$$\mathrm{Type}\;A:\left\{\begin{array}{ll} V_{1}^{1}={\left(\frac{z-z_{1}}{z_{2}-z_{1}}\right)}^{p} & z\in \left[z_{1},z_{2}\right]\\ V_{1}^{2}=1 & z\in \left[z_{2},z_{3}\right]\\ V_{1}^{3}={\left(\frac{z-z_{4}}{z_{3}-z_{4}}\right)}^{p} & z\in \left[z_{3},z_{4}\right] \end{array}\right\}\;\mathrm{Type}\;B:\left\{\begin{array}{ll} V_{1}^{1}=1 & z\in \left[z_{1},z_{2}\right]\\ V_{1}^{2}={\left(\frac{z-z_{3}}{z_{2}-z_{3}}\right)}^{p} & z\in \left[z_{2},z_{3}\right]\\ V_{1}^{3}=0 & z\in \left[z_{3},z_{4}\right] \end{array}\right\}$$

The Voigt criteria can efficiently evaluate the effectiveness of each layer of material, and the Young’s modulus (*E^e^*), Poisson’s ratio (*µ^e^*), and mass density (*ρ^e^*) of the *e*th layer are given using the following equationos:(2a)$$E^{e}=\left(E_{t}^{e}-E_{b}^{e}\right)V_{1}^{e}+E_{b}^{e}$$
(2b)$$\mu^{e}=\left(\mu_{t}^{e}-\mu_{b}^{e}\right)V_{1}^{e}+\mu_{b}^{e}$$
(2c)$$\rho^{e}=\left(\rho_{t}^{e}-\rho_{b}^{e}\right)V_{1}^{e}+\rho_{b}^{e}$$

The subscripts *b* and *t* represent the bottom face and the top face of each layer, respectively. In order to highlight the basic principles of Equations (1) and (2), Figure 3 shows the volume fraction $V_{1}^{e}$ of the FGM sandwich rectangular plate along the thickness direction *z* as a function of the power law index *p*.

### 2.2. Admissible Displacement Functions

The Ritz method is suitable for studying the vibration performance of three-dimensional FGM sandwich rectangular plates under general boundary conditions. Therefore, selecting appropriate displacement admissible functions is critical [58]. It can be found that the conventional displacement admissible functions are usually defined according to the boundary conditions, such as a conventional Fourier series, Chebyshev polynomial, and so on. The latter would cause a tedious formula deduction and reduce computational efficiency. In previous work, the author’s team proposed a new type of displacement admissible function by adding an auxiliary function for the boundary based on the traditional Fourier series [60,61]. In this way, the admissible displacement function and its derivative at the edges of the structure can be solved mechanically. Therefore, an improved Fourier series is used in this study to represent the allowable displacement function of three-dimensional functionally graded sandwich rectangular plates. The specific expressions are as follows [62]:(3a)$$u\left(x,y,z\right)=\left(U^{\Omega }\left(x,y,z\right)+\sum_{q=1}^{6}U_{q}^{S}\left(x,y,z\right)\right)A_{mnq}$$
(3b)$$v\left(x,y,z\right)=\left(V^{\Omega }\left(x,y,z\right)+\sum_{q=1}^{6}V_{q}^{S}\left(x,y,z\right)\right)B_{mnq}$$
(3c)$$w\left(x,y,z\right)=\left(W^{\Omega }\left(x,y,z\right)+\sum_{q=1}^{6}W_{q}^{S}\left(x,y,z\right)\right)C_{mnq}$$
where **U**^Ω^, **V**^Ω^, and **W**^Ω^ refer to the internal displacement distribution functions of the functionally graded sandwich rectangular plate; **U***_q_*^S^, **V***_q_*^S^, and **W***_q_*^S^ are the complementary sequences of boundary displacements of the functionally graded sandwich rectangular plate; and **A***_mnq_*, **B***_mnq_*, and **C***_mnq_* are 3D Fourier coefficient vectors. The following formulas represent the specific parameters of each vector symbol [61]:(4a)$$U^{\Omega }=V^{\Omega }=W^{\Omega }=\left\{\begin{array}{l} \cos\lambda_{0}^{a}x\cos\lambda_{0}^{b}y\cos\lambda_{0}^{h}z,\cdots ,\cos\lambda_{0}^{a}x\cos\lambda_{0}^{b}y\cos\lambda_{Q_{1}}^{h}z,\cdots ,\\ \cos\lambda_{0}^{a}x\cos\lambda_{N_{1}}^{b}y\cos\lambda_{Q_{1}}^{h}z,\cdots ,\cos\lambda_{M_{1}}^{a}x\cos\lambda_{N_{1}}^{b}y\cos\lambda_{Q_{1}}^{h}z \end{array}\right\}$$
(4b)$$U_{1}^{B}=V_{1}^{B}=W_{1}^{B}=\left\{\begin{array}{l} \sin(\lambda_{-2}^{a}x)\cos(\lambda_{0}^{b}y),\cdots ,\sin(\lambda_{-2}^{a}x)\cos(\lambda_{n}^{b}y),\cdots ,\\ \sin(\lambda_{-2}^{a}x)\cos(\lambda_{N}^{b}y),\cdots ,\sin(\lambda_{-1}^{a}x)\cos(\lambda_{N}^{b}y) \end{array}\right\}$$
(4c)$$U_{2}^{B}=V_{2}^{B}=W_{2}^{B}=\Phi_{2}^{B}=\Theta_{2}^{B}=\left\{\begin{array}{l} \cos(\lambda_{0}^{a}x)\sin(\lambda_{-2}^{b}y),\cos(\lambda_{0}^{a}x)\sin(\lambda_{-1}^{b}y),\cdots ,\\ \cos(\lambda_{m}^{a}x)\sin(\lambda_{-2}^{b}y),\cdots ,\cos(\lambda_{M}^{a}x)\sin(\lambda_{-1}^{b}y) \end{array}\right\}$$
(4d)$$U_{1}^{S}=V_{1}^{S}=W_{1}^{S}=\left\{\begin{array}{l} \sin\lambda_{-2}^{a}x\cos\lambda_{0}^{b}y\cos\lambda_{0}^{h}z,\cdots ,\sin\lambda_{-2}^{a}x\cos\lambda_{0}^{b}y\cos\lambda_{Q_{1}}^{h}z,\cdots ,\\ \sin\lambda_{-2}^{a}x\cos\lambda_{N_{1}}^{b}y\cos\lambda_{Q_{1}}^{h}z,\cdots ,\sin\lambda_{-1}^{a}x\cos\lambda_{N_{1}}^{b}y\cos\lambda_{Q_{1}}^{h}z \end{array}\right\}$$
(4e)$$U_{2}^{S}=V_{2}^{S}=W_{2}^{S}=\left\{\begin{array}{l} \cos\lambda_{0}^{a}x\sin\lambda_{-2}^{b}y\cos\lambda_{0}^{h}z,\cdots ,\cos\lambda_{0}^{a}x\sin\lambda_{-2}^{b}y\cos\lambda_{Q_{1}}^{h}z,\cdots ,\\ \cos\lambda_{0}^{a}x\sin\lambda_{-2}^{b}y\cos\lambda_{Q_{1}}^{h}z,\cdots ,\cos\lambda_{M_{1}}^{a}x\sin\lambda_{-1}^{b}y\cos\lambda_{Q_{1}}^{h}z \end{array}\right\}$$
(4f)$$U_{3}^{S}=V_{3}^{S}=W_{3}^{S}=\left\{\begin{array}{l} \cos\lambda_{0}^{a}x\cos\lambda_{0}^{b}y\sin\lambda_{-2}^{h}z,\cos\lambda_{0}^{a}x\cos\lambda_{0}^{b}y\sin\lambda_{-1}^{h}z,\cdots ,\\ \cos\lambda_{0}^{a}x\cos\lambda_{N_{1}}^{b}y\sin\lambda_{-2}^{h}z,\cdots ,\cos\lambda_{M_{1}}^{a}x\cos\lambda_{N_{1}}^{b}y\sin\lambda_{-1}^{h}z \end{array}\right\}$$
(4g)$$U_{4}^{S}=V_{4}^{S}=W_{4}^{S}=\left\{\begin{array}{l} \sin\lambda_{-2}^{a}x\sin\lambda_{-2}^{b}y\cos\lambda_{0}^{h}z,\cdots ,\sin\lambda_{-2}^{a}x\sin\lambda_{-2}^{b}y\cos\lambda_{Q_{1}}^{h}z,\cdots ,\\ \sin\lambda_{-2}^{a}x\sin\lambda_{-1}^{b}y\cos\lambda_{Q_{1}}^{h}z,\cdots ,\sin\lambda_{-1}^{a}x\sin\lambda_{-1}^{b}y\cos\lambda_{Q_{1}}^{h}z \end{array}\right\}$$
(4h)$$U_{5}^{S}=V_{5}^{S}=W_{5}^{S}=\left\{\begin{array}{l} \sin\lambda_{-2}^{a}x\cos\lambda_{0}^{b}y\sin\lambda_{-2}^{h}z,\sin\lambda_{-2}^{a}x\cos\lambda_{0}^{b}y\sin\lambda_{-1}^{h}z,\cdots ,\\ \sin\lambda_{-2}^{a}x\cos\lambda_{N_{1}}^{b}y\sin\lambda_{-2}^{h}z,\cdots ,\sin\lambda_{-1}^{a}x\cos\lambda_{N_{1}}^{b}y\sin\lambda_{-1}^{h}z \end{array}\right\}$$
(4i)$$U_{6}^{S}=V_{6}^{S}=W_{6}^{S}=\left\{\begin{array}{l} \cos\lambda_{0}^{a}x\sin\lambda_{-2}^{b}y\sin\lambda_{-2}^{h}z,\cos\lambda_{0}^{a}x\sin\lambda_{-2}^{b}y\sin\lambda_{-1}^{h}z,\\ \cos\lambda_{0}^{a}x\sin\lambda_{-1}^{b}y\sin\lambda_{-2}^{h}z,\cdots ,\cos\lambda_{M_{1}}^{a}x\sin\lambda_{-1}^{b}y\sin\lambda_{-1}^{h}z \end{array}\right\}$$
(4j)$$\Delta_{m_{1}n_{1}q_{1}}={\left\{\begin{array}{l} \Delta_{0,0,0}^{1},\cdots ,\Delta_{0,0,q_{1}}^{1},\cdots ,\Delta_{0,0,Q_{1}}^{1},\cdots ,\Delta_{0,N_{1},Q_{1}}^{1},\cdots ,\Delta_{m_{1},n_{1},q}^{1},\cdots ,\Delta_{M_{1},N_{1},Q_{1}}^{1},\\ \Delta_{-2,0,0}^{2},\cdots ,\Delta_{-2,0,q_{1}}^{2},\cdots ,\Delta_{-2,0Q_{1}}^{2},\cdots ,\Delta_{-2,n_{1},q_{1}}^{2},\cdots ,\Delta_{-1,n_{1},q_{1}}^{2},\cdots ,\Delta_{-1,N_{1},Q_{1}}^{2},\\ \Delta_{0,-2,0}^{3},\cdots ,\Delta_{0,-2,q_{1}}^{3},\cdots ,\Delta_{0,-2,Q_{1}}^{3},\cdots ,\Delta_{m_{1},-2,q_{1}}^{3},\cdots ,\Delta_{m_{1},-1,q_{1}}^{3},\cdots ,\Delta_{M_{1},-1,Q_{1}}^{3},\\ \Delta_{0,0,-2}^{4},\Delta_{0,0,-1}^{4},\cdots ,\Delta_{0,N_{1},-2}^{4},\cdots ,\Delta_{m_{1},n_{1},-2}^{4},\cdots ,\Delta_{m_{1},n_{1},-1}^{4},\cdots ,\Delta_{M_{1},N_{1},-1}^{4},\\ \Delta_{-2,-2,0}^{5},\cdots ,\Delta_{-2,-2,q_{1}}^{5},\cdots ,\Delta_{-2,-2,Q_{1}}^{5},\cdots ,\Delta_{-2,-1,q_{1}}^{5},\cdots ,\Delta_{-2,-1,Q_{1}}^{5},\cdots ,\Delta_{-1,-1,Q_{1}}^{5},\\ \Delta_{-2,0,-2}^{6},\Delta_{-2,0,-1}^{6},\cdots ,\Delta_{-2,n_{1},-2}^{6},\cdots ,\Delta_{-2,N_{1},-2}^{6},\cdots ,\Delta_{-1,n_{1},Q_{1}}^{6},\cdots ,\Delta_{-1,N_{1},-1}^{6},\\ \Delta_{0,-2,-2}^{7},\Delta_{0,-2,-1}^{7},\cdots ,\Delta_{m_{1},-2,-2}^{7},\cdots ,\Delta_{m_{1},-1,-1}^{7},\cdots ,\Delta_{M_{1},-2,-2}^{7},\cdots ,\Delta_{M_{1},-1,-1}^{7} \end{array}\right\}}^{T}\left(\Delta =A,B,C\right)$$
(4k)$$\lambda_{m_{1}}^{a}=m_{1}\pi /a\lambda_{n_{1}}^{b}=n_{1}\pi /b\lambda_{l_{1}}^{h}=l_{1}\pi /h$$

### 2.3. Energy Expressions

Based on the theory of 3D elasticity [9], the relationship between linear strain and displacement for a three-dimensional functional gradient sandwich rectangular plate are given as:(5)$$\begin{array}{l} \varepsilon_{xx}=\frac{\partial u}{\partial x},\varepsilon_{yy}=\frac{\partial v}{\partial y},\varepsilon_{zz}=\frac{\partial w}{\partial z}\\ \gamma_{xy}=\frac{\partial u}{\partial y}+\frac{\partial v}{\partial x},\gamma_{xz}=\frac{\partial u}{\partial z}+\frac{\partial w}{\partial x},\gamma_{yz}=\frac{\partial v}{\partial z}+\frac{\partial w}{\partial y} \end{array}$$

The stress of the three-dimensional FGM sandwich rectangular plate using the theory of three-dimensional constraint of a linear elastic can be expressed as follows:(6)$$\left\{\begin{array}{c} \sigma_{xx}\\ \sigma_{yy}\\ \sigma_{zz}\\ \sigma_{yz}\\ \sigma_{xz}\\ \sigma_{xy} \end{array}\right\}=\left[\begin{array}{cccccc} C_{11} & C_{12} & C_{13} & 0 & 0 & 0\\ C_{12} & C_{22} & C_{23} & 0 & 0 & 0\\ C_{13} & C_{23} & C_{33} & 0 & 0 & 0\\ 0 & 0 & 0 & C_{44} & 0 & 0\\ 0 & 0 & 0 & 0 & C_{55} & 0\\ 0 & 0 & 0 & 0 & 0 & C_{66} \end{array}\right]\left\{\begin{array}{c} \varepsilon_{xx}\\ \varepsilon_{yy}\\ \varepsilon_{zz}\\ \gamma_{xy}\\ \gamma_{xz}\\ \gamma_{yz} \end{array}\right\}$$
where the stiffness coefficient *C_ij_* is obtained as follows:(7)$$\begin{array}{l} C_{11}=C_{22}=C_{33}=\lambda \left(z\right)+2G\left(z\right)\\ C_{12}=C_{13}=C_{23}=\lambda \left(z\right),\;C_{44}=C_{55}=C_{66}=G\left(z\right)\\ \lambda \left(z\right)=\frac{E(z)}{\left(1+\mu \right)\left(1-2\mu \right)},G\left(z\right)=\frac{E(z)}{2\left(1+\mu \right)} \end{array}$$

Then, based on the elasticity theory, the strain energy **U** for a three-dimensional functionally graded sandwich rectangular plate can be obtained (detailed descriptions of Equation (8) are given in Appendix A).
(8)$$U=\frac{1}{2}\int_{0}^{h}\int_{0}^{a}\int_{0}^{b}\left(\sigma_{xx}\varepsilon_{xx}+\sigma_{yy}\varepsilon_{yy}+\sigma_{zz}\varepsilon_{zz}+\sigma_{xy}\varepsilon_{xy}+\sigma_{xz}\varepsilon_{xz}+\sigma_{yz}\varepsilon_{yz}\right)dxdydz$$

The corresponding kinetic energy (*T*) function of the three-dimensional FGM sandwich rectangular plate can be given as follows:(9)$$T=\frac{1}{2}\int_{0}^{h}\int_{0}^{b}\int_{0}^{a}\rho \omega^{2}\left[{\left(\left(\begin{array}{l} U^{\Omega }+\\ \sum_{q=1}^{6}U_{q}^{S} \end{array}\right)A_{mnq}\right)}^{2}+{\left(\left(\begin{array}{l} V^{\Omega }+\\ \sum_{q=1}^{6}V_{q}^{S} \end{array}\right)B_{mnq}\right)}^{2}+{\left(\left(\begin{array}{l} W^{\Omega }+\\ \sum_{q=1}^{6}W_{q}^{S} \end{array}\right)C_{mnq}\right)}^{2}\right]dxdydz$$

The energy (**U***sp*) kept in reserve by the boundary spring during deformation strain vibrations can be given as follows:(10)$$\begin{array}{l} U_{sp}=\frac{1}{2}\int_{0}^{h}\int_{0}^{b}\left\{\begin{array}{l} k_{x0}^{u}{\left(\left(U^{\Omega }\left(0,y,z\right)+\sum_{q=1}^{6}U_{q}^{S}\left(0,y,z\right)\right)A_{mnq}\right)}^{2}+k_{x1}^{u}{\left(\left(U^{\Omega }\left(a,y,z\right)+\sum_{q=1}^{6}U_{q}^{S}\left(a,y,z\right)\right)A_{mnq}\right)}^{2}+\\ k_{x0}^{v}{\left(\left(V^{\Omega }\left(0,y,z\right)+\sum_{q=1}^{6}V_{q}^{S}\left(0,y,z\right)\right)B_{mnq}\right)}^{2}+k_{x1}^{v}{\left(\left(V^{\Omega }\left(a,y,z\right)+\sum_{q=1}^{6}V_{q}^{S}\left(a,y,z\right)\right)B_{mnq}\right)}^{2}+\\ k_{x0}^{w}{\left(\left(W^{\Omega }\left(0,y,z\right)+\sum_{q=1}^{6}W_{q}^{S}\left(0,y,z\right)\right)C_{mnq}\right)}^{2}+k_{x1}^{w}{\left(\left(W^{\Omega }\left(a,y,z\right)+\sum_{q=1}^{6}W_{q}^{S}\left(a,y,z\right)\right)C_{mnq}\right)}^{2} \end{array}\right\}dydz\\ \;\;\;\;\;\;+\frac{1}{2}\int_{0}^{h}\int_{0}^{a}\left\{\begin{array}{l} k_{y0}^{u}{\left(\left(U^{\Omega }\left(x,0,z\right)+\sum_{q=1}^{6}U_{q}^{S}\left(x,0,z\right)\right)A_{mnq}\right)}^{2}+k_{y1}^{u}{\left(\left(U^{\Omega }\left(x,b,z\right)+\sum_{q=1}^{6}U_{q}^{S}\left(x,b,z\right)\right)A_{mnq}\right)}^{2}+\\ k_{y0}^{v}{\left(\left(V^{\Omega }\left(x,0,z\right)+\sum_{q=1}^{6}V_{q}^{S}\left(x,0,z\right)\right)B_{mnq}\right)}^{2}+k_{y1}^{v}{\left(\left(V^{\Omega }\left(x,b,z\right)+\sum_{q=1}^{6}V_{q}^{S}\left(x,b,z\right)\right)B_{mnq}\right)}^{2}+\\ k_{y0}^{w}{\left(\left(W^{\Omega }\left(x,0,z\right)+\sum_{q=1}^{6}W_{q}^{S}\left(x,0,z\right)\right)C_{mnq}\right)}^{2}+k_{y1}^{w}{\left(\left(W^{\Omega }\left(x,b,z\right)+\sum_{q=1}^{6}W_{q}^{S}\left(x,b,z\right)\right)C_{mnq}\right)}^{2} \end{array}\right\}dxdz \end{array}$$

As stated above, the vibration behavior of the 3D functionally graded sandwich rectangular plate placed on an elastic foundation with a Pasternak type Winkler layer (stiffness *K_w_*) was determined, as well as for the shear layer (stiffness *K_s_*), drawn in Figure 1. The following formula shows the total energy stored in the foundation spring:(11)$$U_{f}=\frac{1}{2}\int_{0}^{a}\int_{0}^{b}\left\{K_{w}{\left(\tilde{W}C_{mn}\right)}^{2}+K_{S}{\left(\frac{\partial \tilde{W}}{\partial x}C_{mn}\right)}^{2}+K_{S}{\left(\frac{\partial \tilde{W}}{\partial y}C_{mn}\right)}^{2}\right\}dxdy$$

The Lagrangian energy function for the three-dimensional functionally graded sandwich rectangular plate can be expressed as:(12)$$L=U+U_{sp}+U_{f}-T$$

### 2.4. Solution Methodology

By applying the Ritz method, the Lagrangian function (**L**) should not be variable, i.e, its variety is equal to zero under time constraint implemented between a fixed primary and final moment of time *t*_0_ and *t*_1_. A homogeneous quadratic function of Fourier series coefficients is given by the Lagrange function **L**. Partial derivatives of the Lagrangian function **L** from Equation (12) with respect to **A***_mnq_*, **B***_mnq_*, and **C***_mnq_* is as follows:(13)$$\frac{\partial L}{\partial A_{mnq}}=\frac{\partial L}{\partial B_{mnq}}=\frac{\partial L}{\partial C_{mnq}}=0$$

The controlling eigenvalues matrix can be obtained using:(14)$$(K-\omega^{2}M)H=0$$
where the parameters of **K** and **M** represent the overall stiffness matrix and the mass matrix, respectively. **H** is the sealed Fourier coefficient vector. The natural frequencies and mode shapes of the three-dimensional FGM sandwich rectangular plate can be obtained by solving Equation (14). Due to space limitations, detailed expressions of matrices **K**, **M**, and **H** are not given here.

## 3. Numerical Results and Discussion

Section 2 established a theoretical model based on the Ritz method. This section highlights numerical discussions based on the above-mentioned Ritz method. The discussion can be divided into three parts: (a) a study of the convergence characteristics of this method, (b) verification of the correctness and efficiency of the current method using a series of numerical examples, and (c) further development of new numerical examples and parametric studies. The material constituents of the functionally graded layers were set as alumina (top surface) and aluminum (bottom surface) in the absence of other provisions. During the course of this research, material properties were defined as follows: aluminum: *E_m_* = 70 GPa, *μ_m_* = 0.3, *ρ_m_* = 2702 kg/m^3^; alumina: *E_m_* = 380 GPa, *μ_m_* = 0.3, *ρ_m_* = 3800 kg/m^3^. For simplicity, the dimensionless natural frequency Ω, equivalent shear ${\bar{K}}_{S}$, and Winkler parameters ${\bar{K}}_{W}$, respectively, are expressed as followings:(15)$$\Omega =\omega b^{2}\sqrt{\rho_{c}/E_{c}}/h,{\bar{K}}_{W}=\frac{K_{W}b^{4}}{D_{m}},{\bar{K}}_{S}=\frac{K_{S}b^{2}}{D_{m}},D_{m}=\frac{E_{m}h^{3}}{12\left(1-\mu^{2}\right)}$$

In addition, in order to further simplify the study, only one group of foundation coefficients $\left({\bar{K}}_{W},{\bar{K}}_{S}\right)$ = (10,10) was adopted for this research. From bottom to top, the proportion of the layer thicknesses was taken to be 1–2–1 in the absence of other provisions.

### 3.1. Convergence Study

In theory, the Ritz method can obtain quite accurate solutions by increasing the number of displacement tolerance function terms. However, due to limited computing hardware resources, too many expansion terms of displacement admissible functions will greatly reduce the computational efficiency of the Ritz method. Convergence properties of the Ritz method under the improved Fourier series were used in this study. Table 1 presents the first six frequency parameters Ω of FGM sandwich plates with diverse truncated numbers of modified Fourier series. The boundary condition of the structure was limited to CCCC (four edges clamped) and SSSS (four edges simply-supported). Furthermore, *b*/*a* = 1 and *h*/*a* = 0.5 represent geometric parameters, and *p* = 1 represents a power law index. Two cases of inelastic and elastic foundations are considered in this case. Table 1 highlights the advantages of the proposed methodology, e.g., fast convergence characteristics and good numerical stability. Compared with the 4 × 4 × 4 terms, the 8 × 8 × 8 terms provided a sufficiently accurate solution. Table 2 shows the convergence of type B FGM sandwich plates with different truncated numbers. Both geometric parameters and the power law exponents were the same as for Table 1. The above-mentioned results exhibited the same trend as the Type A FGM plates. Thus, in the example given below, the admissible function truncated number was chosen as follows: M × N × Q = 8 × 8 × 8.

### 3.2. Validation Studies

The numerical results of the simply supported and clamped FGM sandwich rectangular plates and pure FGM plates are verified in this section. Table 3 presents the first five frequency parameters Ω = *ωb*^2^/*h*(1/10^9^)^1/2^ for a type A FGM sandwich square plate with four edges simply supported, compared with the clamped boundary conditions. Geometric parameters and material parameters were set as follows: *b*/*a* = 1, *h*/*a* = 0.1, and *p* = 1 and 10. The thickness scheme 2–1–2 was adopted in this example. Compared with Table 3, the maximum error of the method given in this paper was less than 1%. Table 4 provides a comparison of frequency parameter Ω for pure FGM plates (type B, 0–1–0) with different power law exponents *p*. Geometric and material parameters were set as follows: *b*/*a* = 1 and *h*/*a* = 0.5. Table 4 shows four kinds of boundary conditions (SSSS, SCSC, SFSF, and SSSF). Based on the above study, four elastic boundary conditions and the classical boundary conditions are described as follow (taking the *x* = 0 edge as an example): clamped (C): *k_u_* = *k_v_* = *k_w_* = 10^15^; free (F): *k_u_* = *k_v_* = *k_w_* = 0; simply-supported (S): *k_u_* = 0, *k_v_* = *k_w_* = 10^15^; elastic restraint 1 (E^1^): *k_u_* = 10^10^, *k_u_* = *k_w_* = 10^15^; elastic restraint 2 (E^2^): *k_v_* = 10^10^, *k_u_* = *k_w_* = 10^15^; elastic restraint 3 (E^3^): *k_w_* =10^10^, *k_u_* = *k_v_* = 10^15^; and elastic restraint 4 (E^4^): *k_u_* = *k_v_* = *k_w_* = 10^10^. Table 3 and Table 4 shows that the present method was verified as being able to handle the study problems.

### 3.3. Parametric Studies

Based on the above study, this section highlights the parametric study of three-dimensional functionally graded sandwich rectangular plates. Figure 4 gives variations of the first three frequency parameters Ω of the 3D FGM sandwich rectangular plate with respect to diverse elastic parameters. The elastic boundary conditions were as follows: the boundary of *x* = *constant* was defined as a fixed boundary; *y* = 0 as a free boundary; *y* = *b* as an elastic boundary; and only one type of elastic stiffness, which varied from 10^4^ to 10^16^, and the remaining type of elastic stiffness was 0. The geometric and material parameters were consistent with Table 1 and Table 2. It can be seen from Figure 4 that the Ω of the plate increased gradually under the trend where the boundary stiffness coefficient increased. When the stiffness exceeded 10^13^, the stiffness coefficient hardly affected the vibration characteristics of the plate. Based on the above, four elastic boundary conditions and the classical boundary conditions were taken to be as follows (taking *x* = 0 edge as an example): clamped (C): *k_u_* = *k_v_* = *k_w_* = 10^15^; free (F): *k_u_* = *k_v_* = *k_w_* = 0; simply supported (S): *k_u_* = 0, *k_v_* = *k_w_* = 10^15^; elastic restraint 1 (E^1^): *k_u_* = 10^10^, *k_u_* = *k_w_* = 10^15^; elastic restraint 2 (E^2^): *k_v_* = 10^10^, *k_u_* = *k_w_* = 10^15^; elastic restraint 3 (E^3^): *k_w_* = 10^10^, *k_u_* = *k_v_* = 10^15^; and elastic restraint 4 (E^4^): *k_u_* = *k_v_* = *k_w_* = 10^10^. Figure 5 and Figure 6 present variations of the first two frequency parameters Ω of type A and type B 3D FGM sandwich rectangular plates with respect to different foundation parameters, where the foundation parameters were changed from 10^−6^ to 10^6^. In Figure 5 and Figure 6, it can be seen that there was an active region for the foundation parameters, which significantly affected the vibration characteristics of the plate, and the influence of the foundation parameters could be neglected outside this region. Figure 7 and Figure 8 show changes in the first two frequency parameters Ω of type A and type B 3D FGM sandwich rectangular plates related to the power law exponent *p*. Different boundary conditions were considered, including two classical and two elastic boundary conditions. It is clear that there was an influence of *p* on vibration characteristics of the plate under different boundary conditions. For SSSS and CCCC, vibration characteristics initially decreased rapidly with the increased gradient coefficient, and then decreased slowly. For the elastic boundary conditions, the impact of *p* on the vibration behavior of the plate became more complex. For example, for the FGM plates with the E^1^E^1^E^1^E^1^ elastic boundary condition, the increased power law exponent caused an increase of the first-order frequency parameters, while second-order frequency parameters were the opposite. Because the vibration of the 3D FGM sandwich rectangular plate was insufficient, new vibration results are given in Table 5 and Table 6 that can provide reference data for a follow-up study. Table 5 and Table 6 show four classical boundaries (CCCC, SSSS, CFCF, and CSCF) and four elastic boundary conditions (E^1^E^1^E^1^E^1^, E^2^E^2^E^2^E^2^, E^3^E^3^E^3^E^3^, and E^4^E^4^E^4^E^4^). It is clear that increasing the thickness of the functionally graded layer reduced the Ω of the 3D FGM sandwich rectangular plate. This phenomenon can be directly seen from the modal shapes, as shown in Figure 9 and Figure 10.

## 4. Conclusions

We studied the vibration characteristics of FGM sandwich rectangular plates on an elastic foundation under general boundary constraints. According to the power-law distribution, it was assumed that the material characteristics vary continuously throughout the thickness range. This paper has studied four common kinds of sandwich FGM plates. The FGM plates’ displacement was represented as the superposition of a Fourier series and an auxiliary polynomial that was used to account for discontinuities of the original displacement function and its related derivatives. In order to overcome the problem of convergence caused by the discontinuity of boundary conditions, the authors have adopted an improved Fourier series based on a traditional Fourier series. The unknown Fourier series expansion coefficients were obtained using the Rayleigh–Ritz method. The technology of an artificial virtual spring could be used to imitate the general boundary constraints of FGM sandwich rectangular plates. It was concluded that the natural characteristics of the FGM sandwich rectangular plate with elastic properties were affected by different boundary conditions, foundation coefficients, material schemes, and elastic restraints. By comparing results existing in the available literature and results obtained by the method used in this paper, the accuracy of the method in predicting the vibration characteristics of FGM sandwich rectangular plate on an elastic foundation was verified. The influence of elastic parameters, foundation parameters, power-law exponents, and layer thickness ratios were studied in detail in this paper, and some important results have been obtained.

## Figures and Tables

**Figure 1 materials-12-03401-f001:**
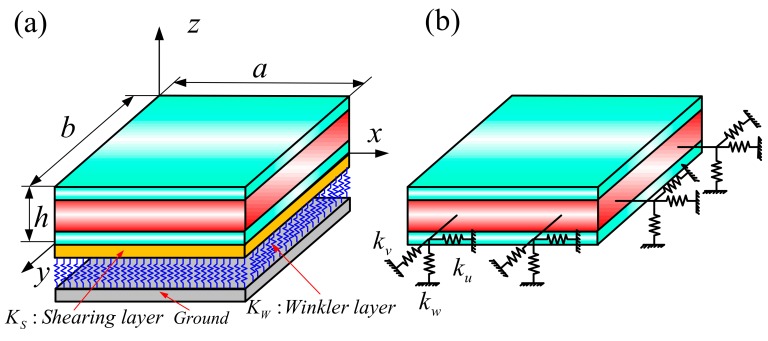
Geometry of a three-dimensional (3D) functionally graded (FG) sandwich rectangular plate: (**a**) the geometry and coordinates, and (**b**) the boundary restraining springs.

**Figure 2 materials-12-03401-f002:**
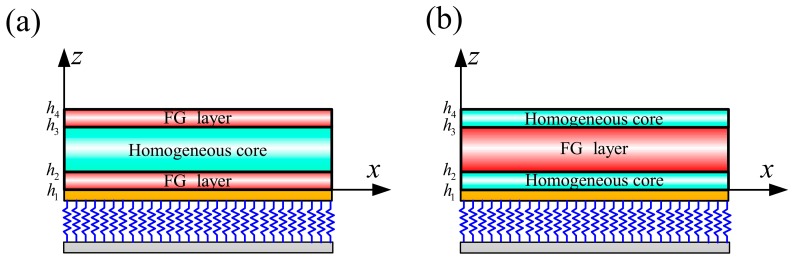
The material variation along the thickness of the 3D FG sandwich rectangular plate: (**a**) FGM facesheet and homogeneous core, and (**b**) homogeneous facesheet and FGM core.

**Figure 3 materials-12-03401-f003:**
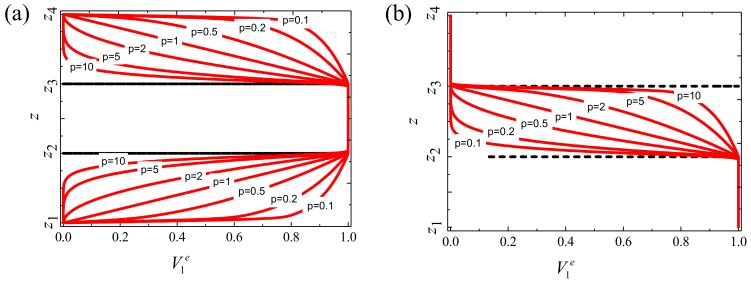
Variation of the volume fraction $V_{1}^{e}$
through the thickness for different values of the power law exponent: (**a**) type A and (**b**) type B.

**Figure 4 materials-12-03401-f004:**
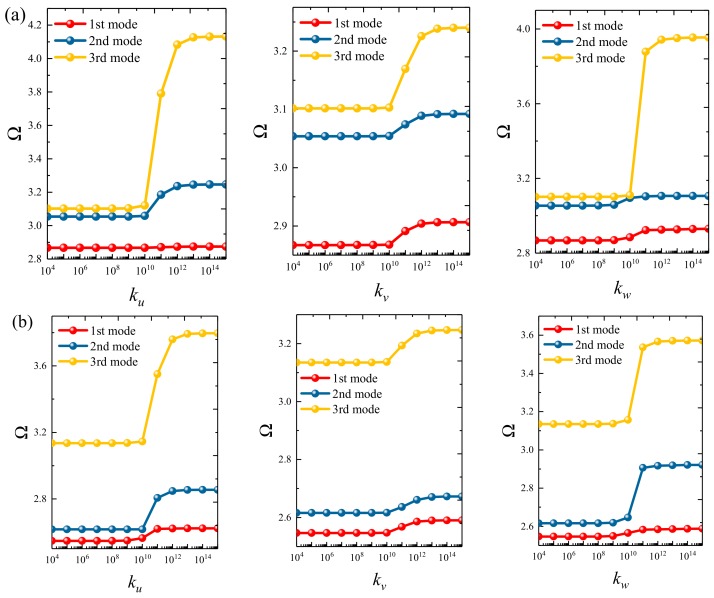
Variation of the non-dimensional frequency parameter Ω of a 3D FG sandwich rectangular plate with respect to different elastic parameters: (**a**) type A and (**b**) type B.

**Figure 5 materials-12-03401-f005:**
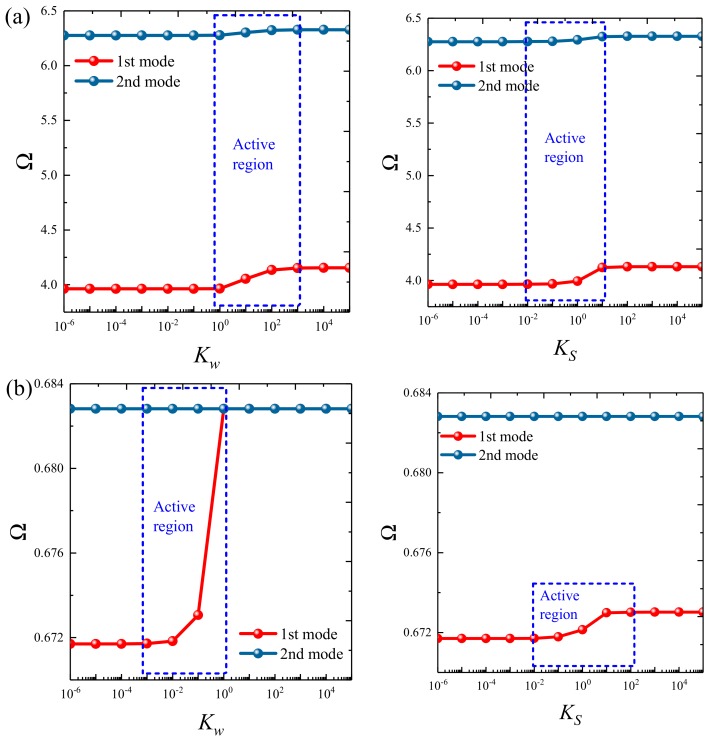
Variation of the non-dimensional frequency parameter Ω of a 3D FG sandwich rectangular plate of type A with respect to different foundation parameters: (**a**) CCCC and (**b**) E4E4E4E4.

**Figure 6 materials-12-03401-f006:**
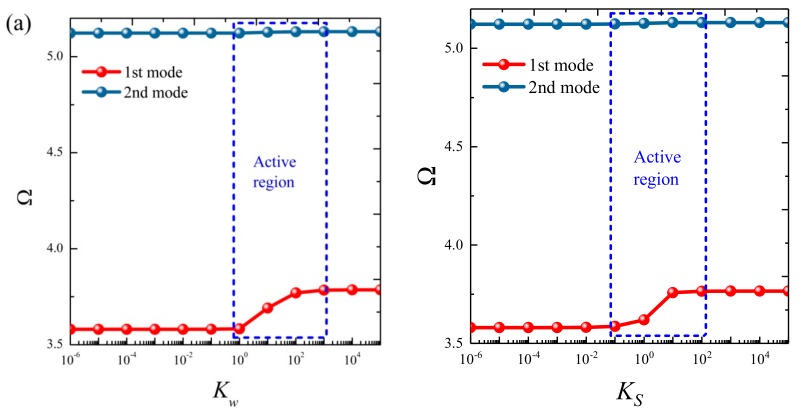
Variation of the non-dimensional frequency parameter Ω of a 3D FG sandwich rectangular plate of type B with respect to different foundation parameters: (**a**) CCCC and (**b**) E4E4E4E4.

**Figure 7 materials-12-03401-f007:**
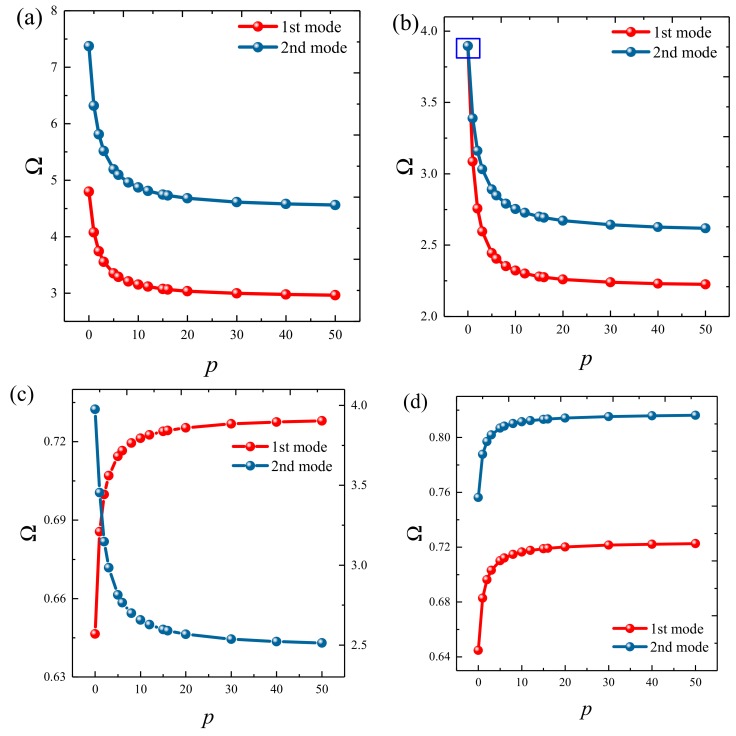
Variation of the non-dimensional frequency parameter Ω of a 3D FG sandwich rectangular plate of type A with respect to different volume fraction indices: (**a**) CCCC, (**b**) SSSS, (**c**) E1E1E1E1, and (**d**) E4E4E4E4.

**Figure 8 materials-12-03401-f008:**
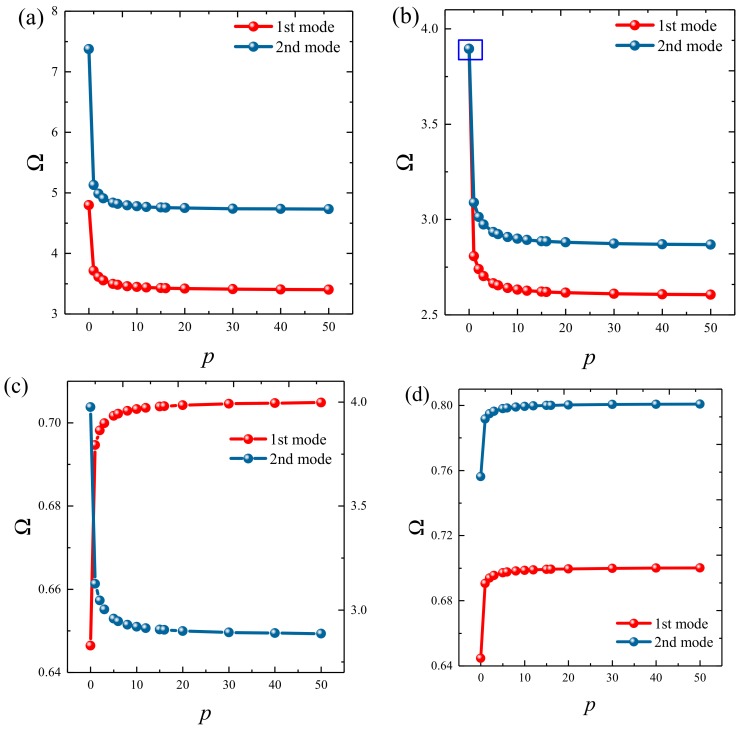
Variation of the non-dimensional frequency parameter Ω of a 3D FG sandwich rectangular plate of type B with respect to different volume fraction indices: (**a**) CCCC, (**b**) SSSS, (**c**) E1E1E1E1, and (**d**) E4E4E4E4.

**Figure 9 materials-12-03401-f009:**
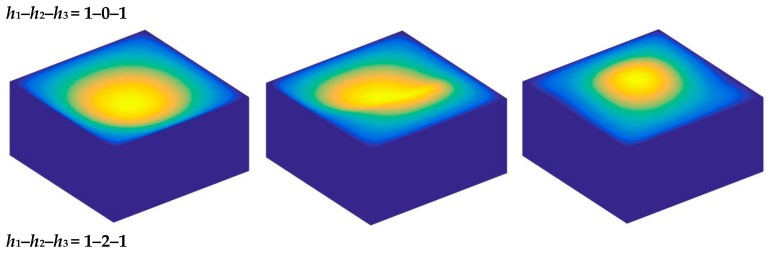
Mode shapes of CCCC 3D FG sandwich rectangular plates of type A with different laminated distribution parameters.

**Figure 10 materials-12-03401-f010:**
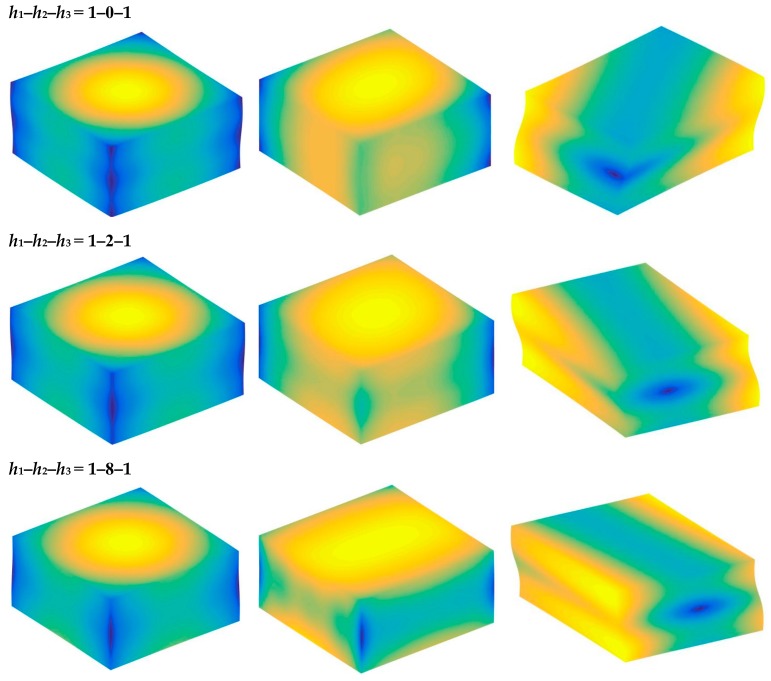
Mode shapes of E4E4E4E4 3D FG sandwich rectangular plates of type B with different laminated distribution parameters.

**Table 1 materials-12-03401-t001:** Convergence of frequency parameters Ω of FGM sandwich plates of type A with different numbers of terms and *p* = 1.

$\left({\bar{K}}_{W},{\bar{K}}_{S}\right)$	M × N × Q	CCCC						SSSS					
		1	2	3	4	5	6	1	2	3	4	5	6
(0,0)	4 × 4 × 4	3.970	6.283	6.283	6.532	6.532	7.637	3.000	3.389	3.389	4.781	5.833	5.833
	5 × 5 × 5	3.965	6.278	6.278	6.528	6.528	7.636	2.998	3.389	3.389	4.781	5.831	5.831
	6 × 6 × 6	3.963	6.277	6.277	6.526	6.526	7.635	2.998	3.389	3.389	4.780	5.831	5.831
	7 × 7 × 7	3.962	6.276	6.276	6.525	6.525	7.635	2.998	3.389	3.389	4.780	5.831	5.831
	8 × 8 × 8	3.961	6.275	6.275	6.524	6.524	7.635	2.998	3.389	3.389	4.780	5.831	5.831
(10,10)	4 × 4 × 4	4.117	6.336	6.336	6.561	6.561	7.638	3.110	3.389	3.389	4.781	5.870	5.870
	5 × 5 × 5	4.105	6.330	6.330	6.555	6.555	7.637	3.104	3.389	3.389	4.781	5.867	5.867
	6 × 6 × 6	4.078	6.318	6.318	6.547	6.547	7.636	3.086	3.389	3.389	4.780	5.858	5.858
	7 × 7 × 7	4.071	6.315	6.315	6.544	6.544	7.635	3.082	3.389	3.389	4.780	5.857	5.857
	8 × 8 × 8	4.052	6.308	6.308	6.540	6.540	7.635	3.069	3.389	3.389	4.780	5.852	5.852

**Table 2 materials-12-03401-t002:** Convergence of frequency parameters Ω of FGM sandwich plates of type B with different numbers of terms and *p* = 1.

$\left({\bar{K}}_{W},{\bar{K}}_{S}\right)$	M × N × Q	CCCC						SSSS					
		1	2	3	4	5	6	1	2	3	4	5	6
(0,0)	4 × 4 × 4	3.590	5.141	5.141	5.823	5.823	5.883	2.736	3.092	3.092	4.151	5.291	5.291
	5 × 5 × 5	3.584	5.125	5.125	5.805	5.805	5.864	2.734	3.089	3.089	4.143	5.283	5.283
	6 × 6 × 6	3.580	5.123	5.123	5.803	5.803	5.864	2.734	3.089	3.089	4.143	5.283	5.283
	7 × 7 × 7	3.575	5.117	5.117	5.792	5.792	5.858	2.732	3.088	3.088	4.140	5.277	5.277
	8 × 8 × 8	3.573	5.115	5.115	5.790	5.790	5.857	2.732	3.088	3.088	4.139	5.277	5.277
(10,10)	4 × 4 × 4	3.755	5.151	5.151	5.884	6.013	6.013	2.826	3.092	3.092	4.151	5.342	5.342
	5 × 5 × 5	3.739	5.132	5.132	5.865	5.992	5.992	2.818	3.089	3.089	4.143	5.336	5.336
	6 × 6 × 6	3.715	5.130	5.130	5.865	5.966	5.966	2.808	3.089	3.089	4.143	5.330	5.330
	7 × 7 × 7	3.701	5.124	5.124	5.859	5.943	5.943	2.802	3.088	3.088	4.140	5.321	5.321
	8 × 8 × 8	3.683	5.122	5.122	5.857	5.919	5.919	2.794	3.088	3.088	4.139	5.314	5.314

**Table 3 materials-12-03401-t003:** Comparison of frequency parameter Ω for type A FGM plates (2–1–2) with a different power law exponent *p*.

B.C	Method	*p* = 1					*p* = 10				
		1	2	3	4	5	1	2	3	4	5
SS SS	Ref. [42,59]	1.30182	3.15875	3.15875	4.91659	6.04048	0.94044	2.28616	2.28616	3.56466	4.38441
	Present	1.30251	3.15931	3.15931	4.91678	6.04735	0.94075	2.28598	2.28598	3.56405	4.38964
	Error (%)	0.05	0.02	0.02	0.00	0.11	0.03	0.01	0.01	0.02	0.12
CC CC	Ref. [42,59]	2.29049	4.46721	4.46721	6.35053	7.56005	1.66075	3.24938	3.24938	4.6307	5.52175
	Present	2.30672	4.50244	4.50244	6.40202	7.61991	1.67303	3.27800	3.27800	4.67538	5.57666
	Error (%)	0.71	0.79	0.79	0.81	0.79	0.74	0.88	0.88	0.96	0.99

B.C: Boundary Conditions.

**Table 4 materials-12-03401-t004:** Comparison of frequency parameter Ω for type B FGM plates (0–1–0) with different a power-law exponent *p*.

*p*	SSSS			SCSC			SFSF			SSSF		
	Ref. [61]	Present	Error (%)	Ref. [61]	Present	Error (%)	Ref. [61]	Present	Error (%)	Ref. [61]	Present	Error (%)
0	1.8470	1.8459	0.06	1.9139	1.9139	0.00	1.0652	1.0645	0.06	0.9570	0.9571	0.01
1	1.4687	1.4687	0.00	1.5724	1.5726	0.01	0.8342	0.8342	0.00	0.7937	0.7940	0.03
2	1.3095	1.3101	0.04	1.4026	1.4031	0.04	0.7464	0.7469	0.06	0.7149	0.7153	0.06
5	1.1450	1.1461	0.09	1.2072	1.2085	0.11	0.6687	0.6694	0.10	0.6168	0.6177	0.15

**Table 5 materials-12-03401-t005:** Frequency parameter Ω of the FGM plates of type A with different boundary conditions.

*h*_1_–*h*_2_–*h*_3_	Mode	Boundary Conditions							
		CCCC	SSSS	CFCF	CSCF	E^1^E^1^E^1^E^1^	E^2^E^2^E^2^E^2^	E^3^E^3^E^3^E^3^	E^4^E^4^E^4^E^4^
	1	3.006	2.277	2.154	2.204	0.735	0.735	0.851	0.729
1–0–1	2	4.579	2.509	2.277	2.400	2.551	2.551	1.930	0.825
	3	4.804	3.523	2.301	2.907	2.667	2.667	2.945	1.034
	1	3.598	2.612	2.529	2.584	0.702	0.702	0.816	0.698
1–1–1	2	5.602	3.109	2.673	2.972	3.017	3.017	2.124	0.796
	3	5.881	4.348	2.843	3.418	3.228	3.228	3.411	0.990
	1	3.906	2.873	2.739	2.800	0.687	0.687	0.800	0.684
1–2–1	2	6.095	3.350	2.902	3.202	3.288	3.288	2.274	0.784
	3	6.398	4.704	3.064	3.746	3.456	3.456	3.682	0.969
	1	4.188	3.172	2.937	3.008	0.673	0.673	0.784	0.670
1–4–1	2	6.532	3.561	3.130	3.405	3.555	3.555	2.447	0.772
	3	6.870	5.023	3.258	4.097	3.657	3.657	3.957	0.950
	1	4.403	3.433	3.091	3.172	0.662	0.662	0.772	0.660
1–8–1	2	6.848	3.707	3.315	3.545	3.771	3.771	2.603	0.763
	3	7.174	5.240	3.393	4.391	3.796	3.796	4.181	0.935

**Table 6 materials-12-03401-t006:** Frequency parameter Ω of the FGM plates of type B with different boundary conditions.

*h*_1_–*h*_2_–*h*_3_	Mode	Boundary Conditions							
		CCCC	SSSS	CFCF	CSCF	E^1^E^1^E^1^E^1^	E^2^E^2^E^2^E^2^	E^3^E^3^E^3^E^3^	E^4^E^4^E^4^E^4^
	1	3.673	2.775	2.521	2.585	0.694	0.694	0.808	0.690
1–0–1	2	5.050	3.068	2.587	2.980	3.094	3.094	2.209	0.791
	3	5.760	4.088	3.120	3.588	3.191	3.191	3.527	0.979
	1	3.399	2.612	2.339	2.405	0.707	0.707	0.822	0.702
1–1–1	2	4.733	2.855	2.408	2.779	2.885	2.885	2.119	0.803
	3	5.442	3.805	2.949	3.382	2.991	2.991	3.317	0.995
	1	3.287	2.561	2.270	2.336	0.713	0.713	0.829	0.708
1–2–1	2	4.617	2.758	2.338	2.689	2.804	2.804	2.090	0.810
	3	5.267	3.696	2.861	3.299	2.900	2.900	3.233	1.004
	1	3.187	2.525	2.212	2.277	0.720	0.720	0.836	0.714
1–4–1	2	4.511	2.662	2.278	2.600	2.734	2.734	2.069	0.817
	3	5.103	3.594	2.768	3.225	2.810	2.810	3.159	1.013
	1	3.116	2.500	2.171	2.237	0.725	0.725	0.842	0.720
1–8–1	2	4.430	2.583	2.237	2.528	2.685	2.685	2.054	0.823
	3	4.977	3.513	2.686	3.168	2.736	2.736	3.106	1.021

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
