# Peer review of "Three-Dimensional Vibration Analysis of a Functionally Graded Sandwich Rectangular Plate Resting on an Elastic Foundation Using a Semi-Analytical Method"

_materials, 2019, doi:10.3390/ma12203401_

Round 1
Reviewer 1 Report
The proposed 3D vibration analysis of functionally graded materials is physically sound and the obtained results could be useful for understanding the mechanical behavior of these materials. However, authors are encouraged to improve their work by taking into account the following remarks:
1) The proposed formalised is based on equations 3, 4, ..., 14, which are not properly justified. So, authors should cite the sources of these equations and describe the validity of them.
2) The comparison of the obtained results with the theoretical and/or experimental ones reported in the literature is highly recommended.
3) The understanding of the manuscript is a bit difficult due its unclear writing, so the English grammar must be improved throughout the manuscript.
Author Response
Reviewer #1
Comments and Suggestions for Authors
The proposed 3D vibration analysis of functionally graded materials is physically sound and the obtained results could be useful for understanding the mechanical behavior of these materials. However, authors are encouraged to improve their work by taking into account the following remarks:
Comment: The proposed formalized is based on equations 3, 4, ..., 14, which are not properly justified. So, authors should cite the sources of these equations and describe the validity of them.
Response: Thank you very much for your valuable suggestion. Three important articles have been cited in the revised manuscript. The specific articles as follows:
Wang, Q., et al., A semi-analytical method for vibration analysis of functionally graded (FG) sandwich doubly-curved panels and shells of revolution. International Journal of Mechanical Sciences, 2017. 134(Supplement C): p. 479-499. Cui, J., Li, Z., Ye, R., Jiang, W., & Tao, S. (2019). A Semianalytical Three-Dimensional Elasticity Solution for Vibrations of Orthotropic Plates with Arbitrary Boundary Conditions. Shock and Vibration, 2019. Zhao, F. Xie, A.-L Wang, et al. “Vibration behavior of the functionally graded porous (FGP) doubly-curved panels and shells of revolution by using a semi-analytical method,” Composites Part B-engineering, vol. 157, pp. 219-238, 2019.
Comment: The comparison of the obtained results with the theoretical and/or experimental ones reported in the literature is highly recommended.
Response: Thank you for your valuable comments on our manuscript. Just as you point, the results have been compared with other published articles in Table 3 and Table 4. The details have been marked in red in the revised manuscript.
Comment: The understanding of the manuscript is a bit difficult due it's unclear writing, so the English grammar must be improved throughout the manuscript.
Response: Authors are sorry for the poorly written English. The manuscript has been edited and proofread by MDPI (English Editing); some non-professional expressions and language errors in grammar and spellings have been rewritten or revised carefully in whole manuscript.

Reviewer 2 Report
The paper submitted to Materials journal is devoted to calculations of vibration of functionally graded sandwich rectangular plates. In last few decades a number of theoretical works have present such type of calculations in different models (ref.1-50), that shows actuality of this research. The novelty of present research is related to taking into account different types of boundary conditions and elastic foundation on which the plate is placed. The paper presents new calculation results and, in principle, they could be of interest for Materials but referee has two important and a lot of not very important critical remarks.
Important remark
1)It is not clear why in Fig.10 (a-d) the fundamental frequencies Ω for n=1 for plates type 1-1 (1-n-1) is lower than for plate type 1-2 (1-n-1). It looks strange because for n>1 the situation is opposite. Why the increase of layer thickness results in change of relation between frequencies? If this is mistake this casts doubt on the calculation results.
2) Why Authors used matrix (6) for elastic constants instead of elastic compliance?
Not important remarks
a) references [18] and [51] duplicate each other;
b) it is not clear what parameters are described by expressions (4b) and 4(c). Parameters U1B, V1B, W1B, and so on are not present in Eqs. ( a,b,c);
c) indexes for \lamda in expressions (4 a-k) should be checked;
d) expression (6) looks strange. Why C_{66} is placed on the place of C_{44}? Also positions of \gamma_{ ij} should be corrected;
e) what means W with tilda in expressions (11) and H in (14)?
f) line 115: what new function has been suggested by Authors and in what work?
g) almost all Tables and Figures do not correspond to the text of article. For example:
*) Table 1, described in the text is absent;
**) text devoted to Table 3 and 4 correspond to Table 2 and 3;
***) Fig.4 – description in text does not correspond to figure caption.
and so on. It is very difficult to show all mistakes.
h) abbreviation used in line 170 and 187 are explained later in lines 199-202;
j) in the text values of elastic stiffness are given without dimension (for example line 195) that looks very strange.
k) conclusions are written in very general form. Novelty and importance of results should be given in very concrete.
In conclusion, the paper may be of interest for Materials but is prepared very carelessly and need considerable revision.
Author Response
Reviewer #2
The paper submitted to Materials journal is devoted to calculations of vibration of functionally graded sandwich rectangular plates. In last few decades a number of theoretical works have present such type of calculations in different models (ref.1-50), that shows actuality of this research. The novelty of present research is related to taking into account different types of boundary conditions and elastic foundation on which the plate is placed. The paper presents new calculation results and, in principle, they could be of interest for Materials but referee has two important and a lot of not very important critical remarks.
Comment: It is not clear why in Fig.10 (a-d) the fundamental frequencies Ω for n=1 for plates type 1-1 (1-n-1) is lower than for plate type 1-2 (1-n-1). It looks strange because for n>1 the situation is opposite. Why the increase of layer thickness results in change of relation between frequencies? If this is mistake this casts doubt on the calculation results.
Response: Thank you very much for your very important and valuable suggestion. All figures have been changed to new ones. The details have been marked in red in the revised manuscript. Authors have added few new figures as well.
Comment: Why Authors used matrix(6) for elastic constants instead of elastic compliance?
Response: Thank you for your suggestion. The elastic constant used in the derivation, is easy to calculate. See ref[61, 62].
Not important remarks
Comment: references [18] and [51] duplicate each other.
Response: Thank you for your careful review on our manuscript, references [51] have been deleted.
Comment: It is not clear what parameters are described by expressions (4b) and 4(c). Parameters U1B, V1B, W1B, and so on are not present in Eqs. ( a,b,c);
Response: We sincerely appreciate for your valuable comments. As can be seen from the previous article, the meaning of these parameters is expressed as U1B, V1B, W1B stand for complementary sequences of boundary displacements of the functionally graded sandwich rectangular plate. Relevant details have been marked in red in the revised manuscript.
Comment: Indexes for \lamda in expressions (4 a-k) should be checked;
Response: All of spelling have been checked, some errors have been corrected. Thank you very much for your valuable suggestion.
Comment: Expression (6) looks strange. Why C_{66} is placed on the place of C_{44}? Also positions of \gamma_{ij} should be corrected.
Response: Thank you very much for your suggestion. Authors have switched the positions of C-{66} and C-{44} and adjusted the positions of omega and gamma. Now those two expressions are corrected.
(Before)
(After)
Comment: what means W with tilda in expressions (11) and H in(14)?
Response: By substituting the corresponding energy expression into Eq.(12) and performing variation operation on the unknown coefficient, the solution equation can be obtained. H is the sealed Fourier coefficient vector. The details have been marked in red in the revised manuscript.
Comment: line 115: what new function has been suggested by Authors and in what work?
Response: The FGM plates displacement was represented as the superposition of a Fourier series and an auxiliary polynomial that is used to account for discontinuities of the original displacement function and its related derivatives. In order to overcome the problem of convergence caused by discontinuity of boundary conditions, authors have adopted an improved Fourier series, being based on traditional Fourier series.
Comment: almost all Tables and Figures do not correspond to the text of article. For example: Table 1, described in the text is absent.
Response: Thank you for your important suggestion. After careful checking, some Tables and Figures have been revised. The details have been marked in red in the revised manuscript.
Comment: Text devoted to Table 3 and 4 correspond to Table 2 and 3;
Response: Thank you for your important suggestion. After careful checking, some Tables and Figures have been revised. The details have been marked in red in the revised manuscript.
Comment: Fig.4 – description in text does not correspond to figure caption.
Response: Thank you for your suggestion. After careful examination, the mistake has been corrected. The details have been marked in red in the revised manuscript.
Comment: Abbreviation used in line 170 and 187 are explained later in lines 199-202.
Response: Thank you for your valuable comments. Authors have added an explanation at the end of line 187.
Comment: In the text values of elastic stiffness are given without dimension (for example line 195) that looks very strange.
Response: Thank you for your valuable comments. According to the available references, only one type of elastic stiffness is available that varies from 104 to 1016, otherwise elastic stiffness is 0. When all boundary stiffness values are set to 0, the free boundary conditions can be adopted.
Comment: Conclusions are written in very general form. Novelty and importance of results should be given in very concrete.
Response: Thank you very much for your valuable advice on this paper. Conclusions been revised to highlight the novelty and importance of this paper. The details have been marked in red in the revised manuscript.
Comment: In conclusion, the paper may be of interest for Materials but is prepared very carelessly and need considerable revision.
Response: Authors are sorry for poorly written English. The manuscript has been edited and carefully checked, many descriptions, Tables and Figures have been revised. The details have been marked in red in the revised manuscript. The manuscript has been edited and proofread by MDPI (English Editing); non-professional expressions and language errors in grammar and spellings have been rewritten or revised carefully in whole manuscript.

Reviewer 3 Report
Factual comments:
I appreciate that the main objective is clearly stated. The results are validated and support the conclusions. It would be interesting for readers to mention a few words about your software like licence, used language, third-part libraries. The computational complexity of the proposed method could be commented on in the paper. The definition of the problem seems to be a bit vague; It is challenging to imagine reproducing the research.Formal comments:
Line 97 is not clear. Are h_1, .., h_4 coordinates? I prefer to present boundary conditions in the mathematical form together with the governing equation. Fig 1. b is not very descriptive. What is the relation between h_i and z_i? Each variable should be defined, including units if it is first used. The form of equations 4.a - 4.j is a little bit unusual the braces are usually used for definition of sets. These equations also take up a lot of space. I would consider moving part of the equations to the annexes. The same could be said about equation (8). Hopefully, it is possible to find some reduced form (at least use of Einstein summation convention). In figures and the text, different fonts for the same symbols are used. It seems that the font used for article text changes (see line 219 or 230). The description of axes in Fig. 8, Fig. 9 is hardly legible.Author Response
Responses to comments on "Three-dimensional vibration analysis of functionally graded sandwich rectangular plate resting on elastic foundation using semi-analytical method" (materials-595533)
Dear Reviewer,
Thank you very much for your review and constructive suggestions for our manuscript “Three-dimensional vibration analysis of functionally graded sandwich rectangular plate resting on elastic foundation using semi-analytical method”(ID: materials- 595533).
We have revised the manuscript to address all your suggestions and comments, and here are our responses to the Reviewer's comments.
Reviewer #3
I appreciate that the main objective is clearly stated. The results are validated and support the conclusions. It would be interesting for readers to mention a few words about your software like license, used language, third-part libraries. The computational complexity of the proposed method could be commented on in the paper. The definition of the problem seems to be a bit vague; It is challenging to imagine reproducing the research.
Comment: Line 97 is not clear. Are h_1, .., h_4 coordinates?
Response: Thank you for your valuable comments. Three elastic layers make up the sandwich plate; h1=0, and h2, h3, and h4 = h represent vertical coordinates from bottom to top, including the two middle interfaces, respectively.
Comment: I prefer to present boundary conditions in the mathematical form together with the governing equation. Fig 1. b is not very descriptive. What is the relation between h_i and z_i?
Response: Thank you very much for your valuable suggestion. The Fig.1(b) has been revised, Fig.1(b) stands for the boundary restraining springs. hi =zi in this paper, we have revised the figure, thank you very much for your careful.
Comment: Each variable should be defined, including units if it is first used. The form of equations 4.a - 4.j is a little bit unusual the braces are usually used for definition of sets. These equations also take up a lot of space. I would consider moving part of the equations to the annexes. The same could be said about equation (8). Hopefully, it is possible to find some reduced form (at least use of Einstein summation convention).
Response: Thank you for your valuable suggestion. Some equations have been moved to annexes, the detailed descriptions of Eq. (8) are given in Appendix. Some other equations are different, it is cannot find reduced form. Thank you very much for your reminder.
Comment: In figures and the text, different fonts for the same symbols are used. It seems that the font used for article text changes (see line 219 or 230). The description of axes in Fig. 8, Fig. 9 is hardly legible.
Response: Thank you for your important suggestion. After careful checking, some Tables and Figures have been revised. Authors are sorry for the poorly written English. The manuscript has been edited and proofread by MDPI (English Editing). The details have been marked in red in the revised manuscript. The Fig.8 and 9 are listed in follow:

Round 2
Reviewer 1 Report
Authors have properly improved their work.
Author Response
Comment: Authors have properly improved their work.
Response: Thank you very much for your encouragement.

Reviewer 2 Report
Authors have made considerable changes in the manuscript. In particular they replaced almost all figures and tables. Despite changes made some questions remains.
Expressions 2a,b,c look wrong (signs before brackets should be change) Line 108. Fig.3 shows variation of volume fraction Vc not V^{e}_{1}. In Table 3 Authors present results of Ref.60. Ref.60 does not contain results corresponding to the case of A-type (2-1-2). In Ref.60 only plate of B-type (0-1-0) is considered. In Table 4 Authors present results of Ref.59. Ref.59 contains results corresponding to the case of A-type, but not for B-type. May be reference [59] should be replaced by [60]? But note that in Ref. [60] the case of h/a = 0.5 is not considered.
According to my mind the paper may be published after minor revision.
Author Response
Comment: Expressions 2a, b, c look wrong (signs before brackets should be change) Line 108. Fig.3 shows variation of volume fraction Vc not V^{e}_{1}.
Response: Thank you very much for your suggestion, the Expressions 2a, b, c may be easy to misunderstand, the expressions have been revised, the details as follows:
Comment: In Table 3 Authors present results of Ref.60. Ref.60 does not contain results corresponding to the case of A-type (2-1-2). In Ref.60 only plate of B-type (0-1-0) is considered. In Table 4 Authors present results of Ref.59. Ref.59 contains results corresponding to the case of A-type, but not for B-type. May be reference [59] should be replaced by [60]? But note that in Ref. [60] the case of h/a = 0.5 is not considered.
Response: Thank you for your careful review on our manuscript. We are so sorry for our carelessness. The reference [60] has been changed to [42, 59] in table 3. The reference [59] has been changed to [60] in table 3.
